# Clustering and reverse transcription of HIV-1 genomes in nuclear niches of macrophages

Elena Rensen[1,2] iD, Florian Mueller[1,*] iD, Viviana Scoca[2] iD, Jyotsana J Parmar[1], Philippe Souque[2] iD, Christophe Zimmer[1,**,†] iD & Francesca Di Nunzio[2,***,†] iD

## Abstract

In order to replicate, human immunodeficiency virus (HIV-1) reverse-transcribes its RNA genome into DNA, which subsequently integrates into host cell chromosomes. These two key events of the viral life cycle are commonly viewed as separate not only in time, but also in cellular space, since reverse transcription (RT) is thought to be completed in the cytoplasm before nuclear import and integration. However, the spatiotemporal organization of the early viral replication cycle in macrophages, the natural non-dividing target cells that constitute reservoirs of HIV-1 and an obstacle to curing AIDS, remains unclear. Here, we demonstrate that infected macrophages display large nuclear foci of viral DNA (vDNA) and viral RNA (vRNA), in which multiple viral genomes cluster together. These clusters form in the absence of chromosomal integration, sequester the paraspeckle protein CPSF6, and localize to nuclear speckles. Surprisingly, these viral RNA clusters consist mostly of genomic, incoming RNA, both in cells where reverse transcription is pharmacologically suppressed and in untreated cells. We demonstrate that following temporary inhibition, reverse transcription can resume in the nucleus and lead to vDNA accumulation in these clusters. We further show that nuclear reverse transcription can result in transcription-competent viral DNA. These findings change our understanding of the early HIV-1 replication cycle and may have implications for addressing HIV-1 persistence.

**Keywords** HIV-1; macrophages; nuclear import; nuclear speckles; reverse transcription

**Subject Categories** Microbiology, Virology & Host Pathogen Interaction

**The EMBO Journal (2021) 40: e105247**

## Introduction

Productive infection by the human immunodeficiency virus 1 (HIV-1), the causative agent of AIDS, requires reverse transcription (RT) of the viral RNA (vRNA) genome into double-stranded viral DNA (vDNA) and subsequent integration of the vDNA into host cell chromosomes. Studies in immortalized cell lines such as HeLa and activated CD4[+] T cells have established a spatiotemporal sequence of events where: (i) vDNA is synthesized by RT in the cytoplasm, with concomitant degradation of the template vRNA, (ii) the vDNA genome translocates into the nucleus, (iii) the integrase enzyme (IN) inserts the vDNA into the genome, and (iv) the integrated vDNA genome undergoes transcription that leads to viral progeny (Freed, 2001; Hu & Hughes, 2012). How the HIV-1 replication cycle proceeds in other cell types remains comparatively underexplored. Indeed, different cell types can exhibit widely divergent responses to viral attacks, owing to, e.g., different restriction factors or immune cell defense mechanisms (Stremlau *et al*, 2004; Goujon *et al*, 2013; Rasaiyaah *et al*, 2013; Lahaye *et al*, 2018). Macrophages are terminally differentiated, non-dividing cells derived from blood monocytes, which play a critical role in the innate and adaptive immune response (Gordon & Taylor, 2005; Koppensteiner *et al*, 2012; Ganor *et al*, 2019). Along with activated CD4[+] T cells, macrophages are natural target cells for HIV-1, and accumulating evidence points to a critical role of these cells in viral persistence, which remains a major roadblock to eradicating HIV (Honeycutt *et al*, 2017; Ganor *et al*, 2019). Despite this importance, the early steps of HIV-1 infection in macrophages remain elusive. Here, we use imaging approaches to visualize and quantify the cellular localizations of vDNA and vRNA in infected macrophages and study their link with RT. Our data reveal that genomic vRNA forms nuclear clusters that associate with nuclear speckle factors and provide surprising evidence that these structures can harbor a nuclear RT activity.

### HIV-1 genomes form large nuclear foci in infected macrophages

To study HIV infection in macrophages, we used primarily THP-1 cells, a human monocytic cell line, and differentiated them into macrophage-like cells by stimulation with phorbol esters (Schwende *et al*, 1996). We infected these cells with VSV-G pseudotyped HIV-1 carrying the HIV-2 accessory protein Vpx (unless stated otherwise), which overcomes the natural resistance of

1 Imaging and Modeling Unit, Institut Pasteur, UMR 3691 CNRS, C3BI USR 3756 IP CNRS, Paris, France
2 Molecular Virology and Vaccinology, Institut Pasteur, Paris, France
*Corresponding author. Tel: +33 01 40 61 31 70; E-mail: fmueller@pasteur.fr
**Corresponding author. Tel: +33 01 40 61 38 91; E-mail: czimmer@pasteur.fr
***Corresponding author. Tel: +33 01 40 61 36 79; E-mail: francesca.di-nunzio@pasteur.fr
†These authors contributed equally to this work

macrophages to viral infection by counteracting the host restriction factor SAMHD1 (Laguette *et al*, 2011). To enable fluorescent labeling of the virus, we used a virus carrying the endogenous viral integrase (IN) gene fused to an HA-tag (Petit *et al*, 1999, 2000). The tagged virus is similarly infectious as the untagged virus (Petit *et al*, 1999; Blanco-Rodriguez *et al*, 2020). We analyzed reverse transcription (RT) with qPCR by measuring vDNA synthesis and the presence of nuclear vDNA forms including 2LTRs at different times post-infection (p.i.). In parallel, we also measured the number of integrated proviruses, revealing a low level of integration in these cells at 24 h p.i. (Appendix Fig S1). These assays indicated that RT peaks at ~ 24 h p.i. and that the formation of episomal nuclear forms (2LTRs), peaks at ~ 30 h p.i., confirming that these early steps of HIV-1 infection in THP-1 macrophage-like cells are delayed relative to HeLa cells (Arfi *et al*, 2008).

To visualize the reverse-transcribed viral DNA genome, we infected THP-1 cells in the presence of the nucleotide analog EdU for 24 h (Peng *et al*, 2014; Stultz *et al*, 2017) with a multiplicity of infection (MOI) of 50 (unless otherwise stated), as measured by qPCR on 293T cells. Fluorescent visualization of EdU was performed at 48 h p.i. by click chemistry, unless stated otherwise. Because the vast majority of THP-1 cells were terminally differentiated, EdU did not incorporate into host cell chromosomes. A minority of cells (~ 5%) exhibited bright EdU signal throughout the nucleus, clearly indicating that these cells failed to differentiate and continued to replicate their DNA (Appendix Fig S2A). We excluded these cells from further analysis throughout this study and only considered the remaining, terminally differentiated cells. Our images revealed strikingly large and bright EdU foci in the nuclei of infected cells, whereas uninfected control cells displayed only a very weak background signal (Fig 1A, Movie EV1, Appendix Fig S2B and C). While the nuclear envelope of some cells displayed invaginations, dual-color imaging of EdU with immunostained lamin in infected cells confirmed that these foci are located within the nuclear lumen (Figs 1C, and EV1A and B). Quantifications indicated that ~ 73% of cells contained one or more foci (Fig 1I) and that foci had a median size (as measured by the full width at half maximum, FWHM, of intensity profiles) of ~ 600 nm (interquartile range ~ 175 nm; $n = 40$ foci) (Fig 1B and F). In addition to these large foci, some infected cells also showed discrete nuclear punctae of much lower brightness, with a size (FWHM) of ~ 370 nm, close to the microscope's theoretical resolution of ~ 300 nm and hence consistent with particles of similar or smaller size (Fig 1A and F). Triple color imaging of EdU together with immunolabeled capsid (CA) and integrase (IN) exhibited clear colocalization, thereby simultaneously confirming the viral nature of EdU-labeled DNA, and showing that the EdU foci (hereafter referred to as "vDNA foci") are enriched in these viral proteins (Fig 1D, Appendix Fig S3). Importantly, although our experiments used viral particles incorporating Vpx to increase HIV-1 infection efficiency, we observed that in absence of Vpx, infected THP-1 cells also form nuclear vDNA foci, ruling out the possibility that these foci are an artifact due to the presence of Vpx (Appendix Fig S4).

We next asked whether the observed vDNA foci might correspond to individual viral genomes by further analyzing their size and dependence on the MOI. First, we considered the possible size of a single ~ 10 Kb long piece of DNA, the approximate length of

the HIV-1 genome. We used polymer simulations that model the chromatinized vDNA as a linear or circular chain of nucleosomes (Arbona *et al*, 2017; Geis & Goff, 2019) (Figs 1E, and EV2A and B). Our simulations predict a distribution of apparent sizes for a single 10 Kb long HIV-1 genome that is close to the microscope's spatial resolution of ~ 300 nm, hence is much smaller than the observed sizes of nuclear vDNA foci (Fig 1F). Thus, nuclear vDNA foci are larger than expected for single genomes, suggesting that they may comprise multiple genomes. Second, we analyzed the intensity of vDNA foci for MOIs of ~ 10, ~ 50, and ~ 100 (Fig 1G and H). Note that EdU labeling does not involve signal amplification, enabling a quantitative interpretation of the measured intensities. If individual foci correspond to individual genomes, their number is expected to increase proportionally with the MOI, but their intensity should not depend on MOI. In our data, both the number of vDNA foci and their peak intensity significantly increased from MOI 10 to MOI 50, and again from MOI 50 to MOI 100 (Fig 1H and I). Therefore, the MOI dependence of the vDNA intensity also argues for the coexistence of multiple genomes within these foci.

## Nuclear vDNA/vRNA foci contain multiple HIV-1 genomes

To analyze the potential association of vDNA foci with vRNA, we used RNA-FISH probes directed against the POL gene of HIV-1 (Tsanov *et al*, 2016). We verified the specificity of RNA-FISH using uninfected cells and ruled out the possibility that RNA-FISH probes bind vDNA using infected cells treated with RNase (Appendix Fig S5A and B). The dual-color EdU/RNA-FISH images showed bright vRNA foci that displayed high and significant colocalization with the vDNA foci (64% of EdU foci contained vRNA, and 30% of vRNA spots contained vDNA; $P < 0.01$), indicating that viral RNA and DNA occupy the same nuclear space (Fig 2A and B; Appendix Fig S6). The vDNA and vRNA intensities in colocalizing spots exhibited a strong positive correlation (Fig 2C).

To directly test for the presence of multiple viral genomes in nuclear foci, we co-infected cells with two HIV-1 strains containing different reporter genes, GFP and luciferase (LUC). We then performed dual-color RNA-FISH with two different sets of probes directed against GFP and LUC (Fig 2D). We confirmed the specificity of our RNA-FISH probes using control cells infected with only one of the two reporter viruses (Appendix Fig S7). In co-infected cells, RNA-FISH revealed strong colocalization of LUC and GFP RNA (45% of LUC RNA foci contained GFP RNA, and 35% of GFP RNA foci contained LUC RNA; $P < 0.01$) (Fig 2E). In colocalizing foci, intensities of GFP and LUC RNA correlated positively (Fig 2F). Taken together with our observation that vRNA foci occupy the same nuclear regions as vDNA foci (Fig 2A and B), these data firmly establish that nuclear foci are clusters consisting of multiple HIV-1 genomes.

## Viral genome clusters are associated with nuclear body factors

Inspection of our dual EdU/DAPI images indicates that vDNA clusters are located in nuclear regions with lower densities of host cell DNA (Appendix Fig S8; Fig 3E). This opens the possibility of an association of HIV genomes with nuclear bodies (Matera, 1999; Spector, 2001), membrane-less compartments located in the

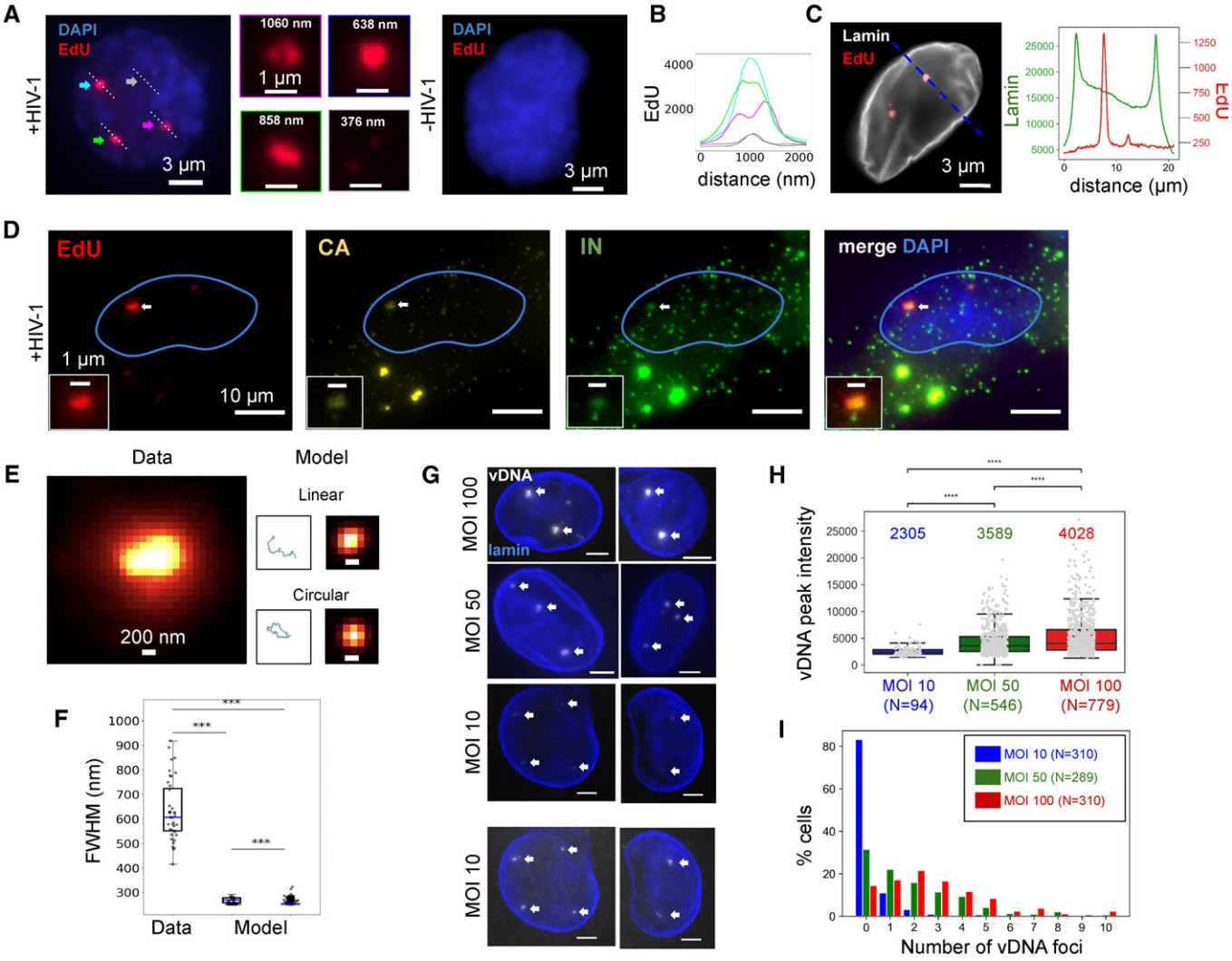

**Figure 1.  HIV genomes form large nuclear foci in macrophages.**

A–C   EdU-labeled viral DNA forms large foci in THP-1 cells infected by HIV-1. EdU images are shown in red, DAPI images in blue. (A) Left: nucleus of an infected cell at 48 h p.i. displays three large and bright EdU foci (colored arrows), as well as small and dim punctae (gray arrows). Center: Magnified views have colored borders matching the colors of the arrows. 3D images show that EdU foci are located within the nucleoplasm (Movie EV1, Appendix Fig S2). Right: nucleus of an uninfected cell displays no EdU signal. (B) Graph shows EdU intensity profiles along lines shown in (A), with colors matching the corresponding arrows. (C) Dual-color image of an infected cell with EdU (red) and immunolabeled lamin (gray). Curves to the right plot the intensity profile of EdU and lamin along the dashed blue line. The EdU intensity peaks do not coincide with lamin enrichment. See also Fig EV1A and B.

D     Multicolor images of an infected cell showing EdU (red) with CA (yellow) and integrase (green). The colocalization of nuclear EdU foci with CA and IN (arrows) confirms that EdU specifically labels viral DNA and shows the presence of these proteins in vDNA foci. See also Appendix Fig S3.

E, F   vDNA foci are much larger than the predicted size of single viral genomes. (E) Left image shows an observed, EdU-labeled vDNA focus. Right: simulations of a single linear or circular 10 Kb long chromatinized DNA polymer chain and corresponding predicted images in diffraction-limited (~ 300 nm resolution) microscopy. (F) Boxplots show the distribution of sizes (FWHM, full width at half maximum) of $n = 40$ measured vDNA foci (for MOI 100) compared to the sizes predicted for linear (left) and circular (right) chains based on $n = 100$ simulated configurations each. Blue lines in boxes define medians, top and bottom limits define upper and lower quartiles, respectively. Whiskers show the full data range, except for outliers. Gray dots are individual data points. All differences are highly significant (Wilcoxon test data vs. model: ***$P \approx 3 \times 10^{-20}$, circular vs. linear model: ***$P \approx 10^{-17}$).

G     Images of vDNA foci (arrows) in THP-1 cells infected with multiplicities of infection (MOI) 10, 50, and 100. Images for MOI 10 are shown at two different contrast levels to reveal dimmer spots. Scale bars, 3 μm.

H     Boxplots compare the peak intensities inside individual vDNA foci for MOIs 10, 50, and 100. Central lines in boxes define medians, top and bottom limits define upper and lower quartiles, respectively. Whiskers show the full data range, except for outliers. Gray dots are individual data points. Intensities increase significantly with MOI (Wilcoxon tests: MOI 10 vs. MOI 50: ***$P \approx 5 \times 10^{-16}$; MOI 50 vs. MOI 100: ***$P \approx 5 \times 10^{-5}$; MOI 10 vs. MOI 100: ***$P \approx 2 \times 10^{-23}$). Data from one experiment. A replicate showed similar results.

I     Histograms show the number of foci per nucleus for MOIs of 10 (blue), 50 (green), and 100 (red).

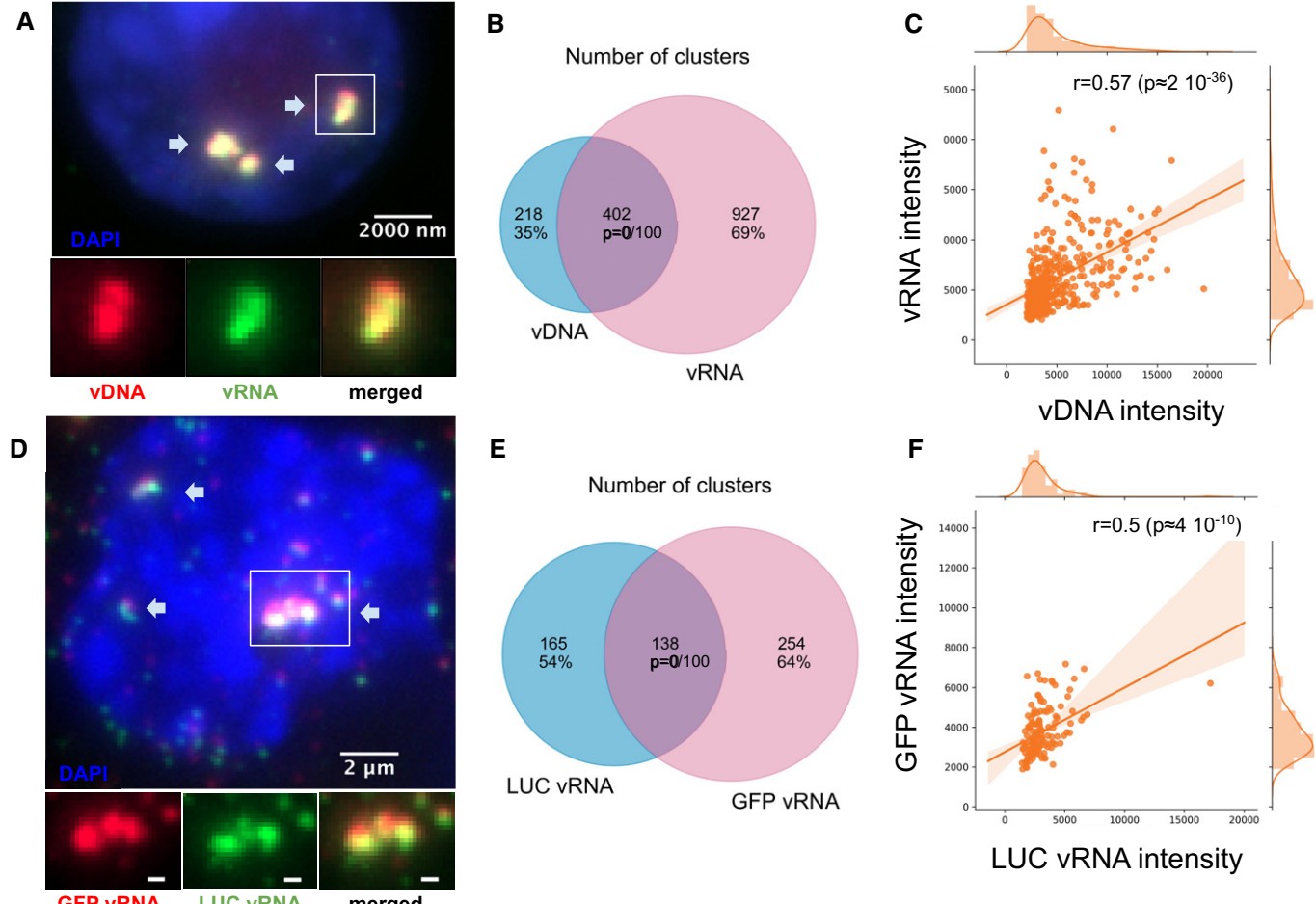

**Figure 2. Nuclear foci contain multiple viral RNAs and DNA genomes.**

A–C   vDNA foci are also foci of vRNA. (A) Dual-color image of an infected THP-1 cell showing the viral DNA (red) and the RNA visualized by RNA-FISH (green). Arrows show foci containing both vDNA and vRNA. The bottom row shows the boxed region magnified, with vDNA and vRNA displayed separately and in combination. See also Appendix Fig S6. (B) Venn diagram shows the number of vDNA foci and vRNA foci and the number of vDNA foci colocalizing with vRNA foci. The *P*-value indicates the significance of colocalization based on a jittering analysis (see Materials and Methods). (C) Scatter plot shows the intensities of vDNA and vRNA in colocalizing foci with the Spearman correlation *r* and associated *P*-value.

D–F   In THP-1 cells co-infected with a HIV1-GFP virus and a HIV1-LUC virus, nuclear foci contain mixtures of both viruses. (D) Dual-color image shows RNA-FISH against GFP (red) and RNA-FISH against LUC (green). Arrows show foci containing both GFP vRNA and LUC vRNA. The bottom row shows the boxed region magnified, with GFP vRNA and LUC vRNA displayed separately and in combination. See Appendix Fig S7. (E) Venn diagram shows the number of GFP RNA foci, the number of LUC RNA foci, and the number of GFP RNA foci colocalizing with LUC RNA foci. (F) Scatter plot shows intensities of GFP RNA and LUC RNA in colocalizing foci with the Spearman correlation *r* and associated *P*-value.

interchromosomal space. We therefore examined the possible association of viral clusters with host proteins, starting with the cleavage and polyadenylation-specific factor subunit 6 (CPSF6). CPSF6 is known to interact with CA and has been implicated in the regulation of different steps of the viral life cycle from HIV-1 nuclear import to integration site distribution (Lee *et al*, 2010; Price *et al*, 2012; Achuthan *et al*, 2018; Buffone *et al*, 2018; Burdick *et al*, 2020). In uninfected cells, CPSF6 displayed a diffuse nucleoplasmic signal; in infected cells, however, CPSF6 accumulated in a small number of nuclear foci, which colocalized with vDNA, in agreement with a recent report (Bejarano *et al*, 2019) (median Pearson's $r = 0.30$; Costes $P < 0.01$ for 16 out of 19 regions of interest) (Fig 3A and D; Appendix Fig S9). CPSF6 is known to associate with paraspeckles,

nuclear bodies often found in the vicinity of nucleoli and speckles (Fox *et al*, 2002; Naganuma & Hirose, 2013). To test whether viral clusters associate with paraspeckles, we imaged the non-coding RNA NEAT1 (Nuclear Enriched Abundant Transcript 1/Human nuclear paraspeckle assembly transcript 1), an essential architectural component of paraspeckles (Naganuma & Hirose, 2013; Zhang *et al*, 2013; Yamazaki *et al*, 2018). In uninfected cells, NEAT1 was enriched in a small number of nuclear clusters, as expected for paraspeckles (Fig 3B; Appendix Fig S10A). In infected cells, NEAT1 displayed a similar localization pattern, and NEAT1 foci appeared in close vicinity to the vDNA clusters, but did not overlap with them and appeared to be excluded (median Pearson's $r = -0.19$; Costes $P < 0.01$ for 19/19 regions of interest)

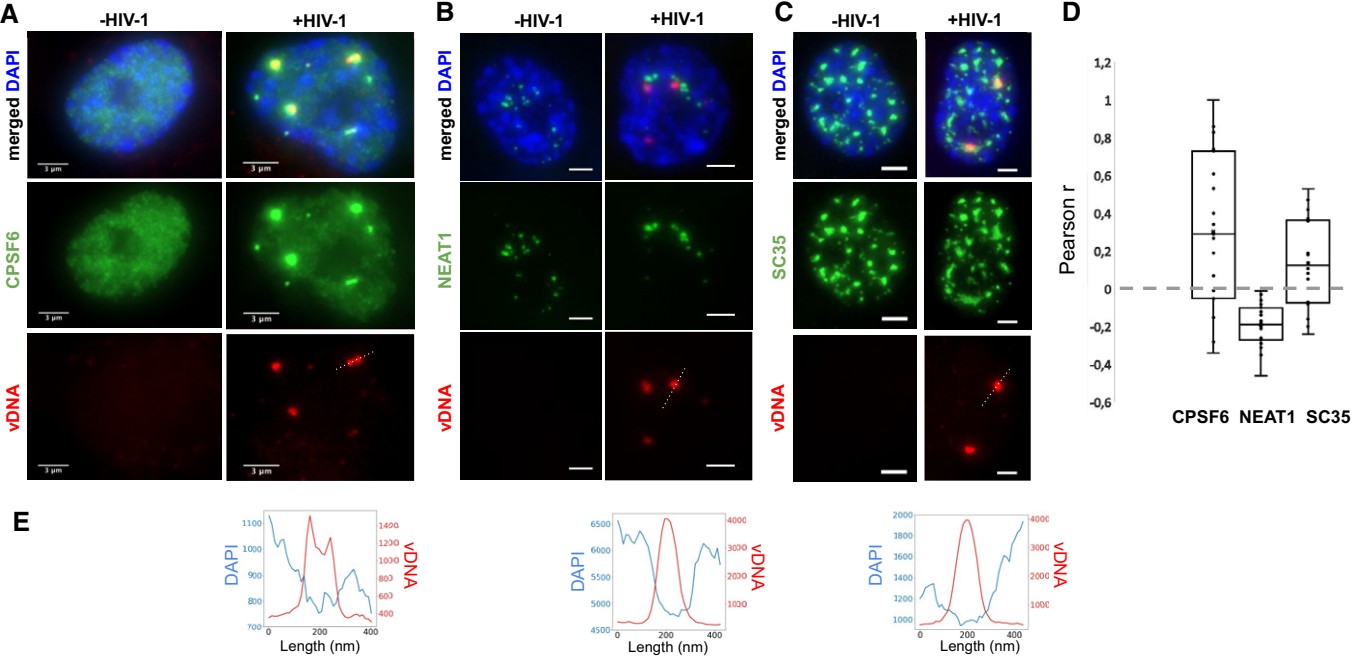

**Figure 3. Viral clusters contain specific nuclear body factors.**

A–C    Images of infected (right) or uninfected (left) THP-1 cells showing the vDNA (EdU) in red, the nucleus (DAPI) in blue, and selected nuclear body factors in green. (A) Green image shows immunolabeling of CPSF6. (B) Green image shows RNA-FISH against NEAT1. (C) Green image shows immunolabeling of SC35. See also Appendix Figs S9–S14.

D       Boxplots show Pearson correlations *r* between vDNA (EdU) and CPSF6, NEAT1, or SC35 in 17–19 regions of interest (ROIs). Central lines in boxes define medians, top and bottom limits define upper and lower quartiles, respectively. Whiskers show the full data range, except for outliers. Gray dots are individual data points. Significance of positive or negative correlations was assessed using the Costes method of random ROI shifts. Highly significant ($P < 0.01$) positive correlations between vDNA and CPSF6 intensities are found in 16 out of 19 ROIs; highly significant negative correlations between vDNA and NEAT1 are found in 19/19 ROIs, and highly significant positive correlations between vDNA and SC35 are found for 16/17 ROIs. Data are from one experiment. For CPSF6, the experiment was repeated with similar results.

E       Intensity profiles of EdU and DAPI along the dotted lines in the vDNA images above.

---

(Fig 3B and D; Appendix Fig S10B). The core paraspeckle protein NONO (p54nrb), recently found to be essential for triggering an immune response to HIV in dendritic cells through interaction with CA (Lahaye *et al*, 2018), was also excluded from vDNA (Fig EV3A–D). Thus, viral clusters do not localize to paraspeckles, but upon infection, the paraspeckle factor CPSF6 relocates to viral clusters.

Paraspeckles are often found in proximity to speckles, nuclear bodies enriched in pre-mRNA splicing factors (Spector & Lamond, 2011). Therefore, our data raised the possibility that viral clusters are associated with speckles. To investigate this, we imaged the non-small nuclear ribonucleoprotein particle factor SC35, a well-studied splicing regulator, and bona fide marker of speckles (Spector & Lamond, 2011). Images of SC35 in uninfected cells showed a large number (typically ∼ 10–15) large nuclear bodies (typical size ∼ 1–2 μm), as expected for speckles (Lamond & Spector, 2003) (Fig 3C). In HIV-1 infected cells, SC35 displayed a similar pattern. Interestingly, nuclear vDNA clusters colocalized with a subset of these SC35 positive bodies (median Pearson's *r* = 0.14; Costes *P* < 0.01 in 16/17 regions) (Fig 3C and D; Appendix Fig S11). Thus, our data suggest that HIV-1 genome clusters enriched in the paraspeckle protein CPSF6 associate with the splicing and speckle factor SC35.

**Viral integration is not required for viral DNA cluster formation**

To determine whether the formation of vDNA clusters requires the integration of the viral genome into host chromosomes, we infected cells with an integration-deficient mutant virus (D116A) (Berger *et al*, 2009). Interestingly, we again observed the presence of large vDNA clusters which are also associated with vRNA (Fig 4A, Appendix Fig S12). As for the virus containing the functional integrase (Fig 2A–C), the colocalization of vDNA and vRNA was highly significant and intensities in colocalizing clusters correlated significantly (Spearman's $r = 0.51$, $P = 1.6 \times 10^{-5}$) (Fig 4B and C). Additionally, when we treated cells at the time of infection with 10 μM of Raltegravir (RAL), an inhibitor of HIV integration, we also observed large and bright vDNA clusters associated with vRNA (Appendix Fig S13B). These experiments demonstrate that the formation of vDNA/vRNA clusters does not depend on integration of the viral genome into host chromosomes.

**Viral RNA clusters contain mostly genomic RNA**

The dual-color EdU/RNA-FISH images reported above clearly indicate that vDNA clusters colocalize with vRNA (Fig 2A–C). However, our RNA-FISH probes against the HIV-1 POL gene cannot

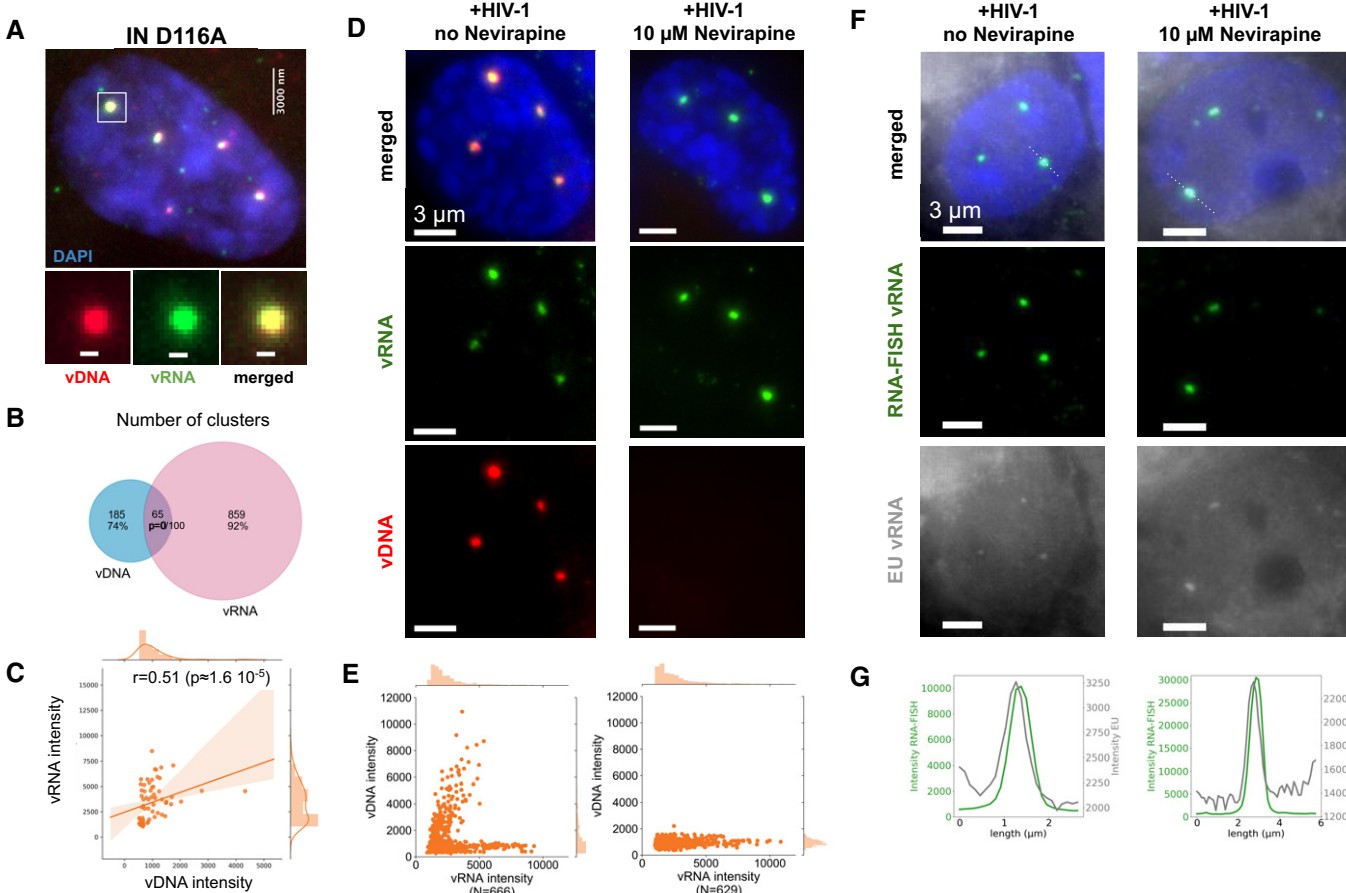

**Figure 4. Nuclear clusters can form in absence of integration and contain incoming vRNA.**

A   Image of a THP-1 cell infected with an integration-deficient HIV-1 carrying the mutation D116A in the catalytic site of IN. The EdU-labeled vDNA is shown in red and the vRNA detected by RNA-FISH in green. The nucleus (DAPI) is shown in blue. See Appendix Fig S12 for a larger image region.

B   Venn diagram shows the number of vRNA clusters, the number of vDNA clusters, and the number of vRNA clusters colocalizing with vDNA clusters. The *P*-value indicates the significance of colocalization based on a jittering analysis (see Materials and Methods).

C   Scatter plot shows vDNA and vRNA intensities in colocalizing vDNA and vRNA clusters with the Spearman correlation *r* and associated *P*-value.

D   THP-1 cells were infected with HIV-1 at an MOI of 20, in absence (left) or presence (right) of nevirapine (NVP, 10 μM). Blue: DAPI staining. Green: vRNA (RNA-FISH). Red: vDNA (EdU).

E   Scatter plots show intensities of vDNA and vRNA in detected vRNA clusters.

F   Image of a THP-1 cell infected by HIV-1 with EU-labeled RNA in absence (left) or presence (right) of nevirapine (NVP, 10 μM). Blue: DAPI staining. Green: vRNA (RNA-FISH). Gray: EU-labeled vRNA. See Appendix Fig S19 for a larger image region.

G   Intensity profiles of EU and RNA-FISH along the dotted lines in panel (F) (top).

distinguish between incoming (genomic) vRNA and newly transcribed (messenger) vRNA. We therefore sought to verify the nature of the observed vRNA by treating cells with 10 μM of nevirapine (NVP), a potent RT inhibitor. The efficacy of the drug was demonstrated by the absence of detectable vDNA in infected nuclei; to our surprise, however, we still observed bright vRNA clusters in the nucleus, whether we exposed the cells to NVP for 24 h, 48 h, or 72 h (Fig 4D and E; Appendix Figs S1, S14A and S20). Because HIV-1 transcripts cannot be produced in the absence of vDNA, we conclude that nuclear vRNA clusters at these time points contain mostly incoming, genomic vRNA, in NVP treated cells.

We next asked if genomic vRNA clusters also exist in absence of RT inhibition. Intensity distributions of vRNA clusters in NVP treated cells at 48 h p.i. were similar to those in untreated cells

(Appendix Fig S14B), strongly suggesting that the vRNA clusters in the untreated cells are also mostly composed of genomic vRNA. To verify this directly, we turned to a different labeling technique that, unlike RNA-FISH, highlights the genomic, but not the transcribed, vRNA. Specifically, we used 5-ethynyl uridine (EU), another nucleoside analog that incorporates into nascent RNA during transcription and can be fluorescently detected by click chemistry, as for EdU (Jao & Salic, 2008; Xu *et al*, 2013). As demonstrated previously (Xu *et al*, 2013), we used EU to label HIV RNA in 293T producer cells, then we infected NVP treated or untreated THP-1 cells with this virus (MOI 50), and performed fixation at 24 h p.i., followed by click chemistry labeling and fluorescence imaging, in combination with RNA-FISH (Fig 4F, Appendix Fig S15). Despite the presence of a significant background signal, we observed EU spots in the

nucleus of both NVP treated and untreated cells that coincided with vRNA clusters (Fig 4G, Appendix Fig S15). These data therefore strongly support the presence of genomic vRNA in the nucleus, regardless of whether cells have been pharmacologically treated.

## Nuclear viral RNA clusters can undergo reverse transcription

The unexpected presence of genomic vRNA clusters in the nucleus (Fig 4D–G) raises the intriguing question as to whether they can serve as templates for RT. To test this possibility, we took advantage of the reversibility of RT inhibition by NVP. We reasoned that if RT occurs locally in vRNA clusters, allowing RT to resume after temporary RT inhibition would lead to the detection of newly synthesized vDNA within these nuclear structures.

We first exposed cells to NVP starting from the time of infection during 24, 48, or 72 h. Dual-color images obtained at all three time points showed an absence of nuclear vDNA signal, as expected, while the vRNA clusters remained clearly visible, as above (Appendix Fig S14A, Fig 4D). Next, we exposed cells to NVP starting from the time of infection for 48 h or 72 h, then removed NVP by wash-out and imaged cells 24 h later. Strikingly, in both experiments, we observed clear vDNA clusters that colocalized with the vRNA clusters, and displayed positive correlation of their intensities (Fig 5A, B, E and F; Appendix Fig S16), as in untreated cells (Fig 2A–C). This appearance of vDNA at vRNA clusters can be explained by a local RT activity within the nuclear vRNA clusters. Alternatively, the observed vDNA clusters might reflect nuclear import of vDNA synthesized exclusively in the cytoplasm, between the time points of NVP wash-out and fixation. In the first case (nuclear RT), we expect that clusters containing more vRNA also exhibit higher amounts of vDNA. This is indeed the case, as evidenced by the positive correlations between vRNA and vDNA intensities in colocalizing clusters in all wash-out experiments (Fig 5E).

To further assess both possibilities, we next quantified the amount of vRNA in the cytoplasmic and nuclear compartments using an automated analysis based on FISH-quant and ImJoy (Mueller et al, 2013; Ouyang et al, 2019). These analyses indicated that the nuclear vRNA pool exceeded the cytoplasmic pool by at least ~ 7-fold after 72 h of NVP treatment (Appendix Fig S14C). Thus, the amount of vRNA available for cytoplasmic RT and subsequent nuclear import during the 24 h after wash-out is much smaller than the amount of vRNA available for nuclear RT. Therefore, the nuclear vDNA detected in our NVP wash-out experiments more likely arises from local RT in nuclear clusters rather than import of vDNA synthesized in the cytoplasm.

To directly rule out a contribution of cytoplasmically synthesized vDNA to nuclear vDNA clusters after NVP removal, we aimed to pharmacologically block nuclear import using the molecule PF-3450074 (PF74). At low concentrations (1.25–2.5 μM), PF74 impedes nuclear import of HIV-1 without abolishing RT (Blair et al, 2010; Francis & Melikyan, 2018; Bejarano et al, 2019; Balasubramaniam et al, 2019; Blanco-Rodriguez et al, 2020). Indeed, when we exposed cells to 1.5 μM of PF74 for 72 h after infection, qPCR indicated that RT was roughly halved, while 2LTR formation was virtually eliminated (Fig 5G). We observed no discernable nuclear vRNA in PF74-treated cells at 48 h or 72 h p.i. and a higher cytoplasmic vRNA pool as compared to NVP treated cells at

the same time point, confirming inhibition of nuclear import (Fig 5C, Appendix Figs S17A and B, and S18). Finally, we treated cells with NVP for 24 h p.i., then washed NVP out and immediately exposed the cells to 1.5 μM PF74 for another 24 h. Despite the block of nuclear import, we again observed clear vDNA clusters colocalizing with vRNA clusters in the nucleus, with positively correlating intensities (Fig 5D and H; Appendix Fig S17C). We made the same observation when exposing cells to NVP for 48 h, applying PF74 at 36 h, and washing out NVP (while keeping the cells exposed to 1.5 μM of PF74) at 48 h p.i. (Appendix Fig S19). We additionally performed qPCR analyses of circular 2LTRs, an exclusively nuclear form of vDNA that can persist in the nucleus for several weeks (Gillim-Ross et al, 2005) (Appendix Fig S1). Interestingly, we were able to amplify 2LTRs in samples after NVP wash-out, whether or not followed by exposure to PF74, albeit with a 3–7 fold reduction compared to untreated cells; this reduction was ~ 10-fold when applying PF74 at 36 h p.i. and washing out NVP at 48 h p.i. (Appendix Fig S20). Nevertheless, the detection of 2LTRs in these experiments corroborates the synthesis of complete vDNA by RT in the nucleus. We also analyzed the presence of proviral DNA using ALU-PCR (Di Nunzio et al, 2013; Lelek et al, 2015). While we could amplify these integrated vDNA forms in untreated cells or in simple NVP wash-out experiments, we failed to do so in the experiments involving PF74 exposure (Appendix Fig S20). This suggests that the majority of the vDNA detected consist of unintegrated genomes, in agreement with our above findings using an integration-deficient virus (Fig 4A–C) or when inhibiting integration pharmacologically (Appendix Fig S12). Notwithstanding, our data argue against the possibility that the nuclear vDNA clusters arise exclusively from import of cytoplasmically synthesized vDNA and therefore constitute compelling evidence for a nuclear RT activity within vRNA clusters after the release of RT inhibition.

## Nuclear RT activity results in transcription-competent vDNA

In addition, we asked whether the viral DNA synthesized in the nucleus after pharmacological block followed by release of RT is competent for transcription. To address this, we infected cells with the virus carrying the GFP reporter gene used above (Fig 2D, Appendix Fig S21) and quantified the percentage of GFP-positive cells using flow cytometry (FACS) (Fig 6A–D, Appendix S21G and H). Cells infected with MOI 20 and fixed at 3 (Appendix Fig S21G) days or 7 days (Fig 6B) p.i. displayed ~ 9 and ~ 12% of GFP-positive cells, respectively, vs. 0% for uninfected cells (Fig 6A). When we inhibited RT by NVP treatment for 3 days, allowed RT to resume by washing out NVP and fixed cells for imaging 4 days later, ~ 4% of cells were found to be GFP positive (Fig 6C) against 0.3% of GFP+ cells in presence of NVP at 7 days p.i. (Appendix Fig S21H). The percentage of GFP-positive cells was lower when 1.5 μM of PF74 was added at the moment of NVP wash-out, or 12 h before the wash-out, to block nuclear import of the remaining cytoplasmic virus (~ 1.2 and ~ 0.94%, respectively) (Fig 6D, Appendix Fig S21H). Although these percentages are considerably lower than for untreated cells, they are much larger than for cells treated with NVP until fixation (0.04% at 3 days p.i., 0.33% at 7 days p.i.) (Appendix Fig S21G and H) or cells treated with PF74 for 3 days (0.15%) (Fig S21G). These FACS results were corroborated by imaging (Fig 6E–H, Appendix Fig S21A). Importantly, we did not use EdU in these

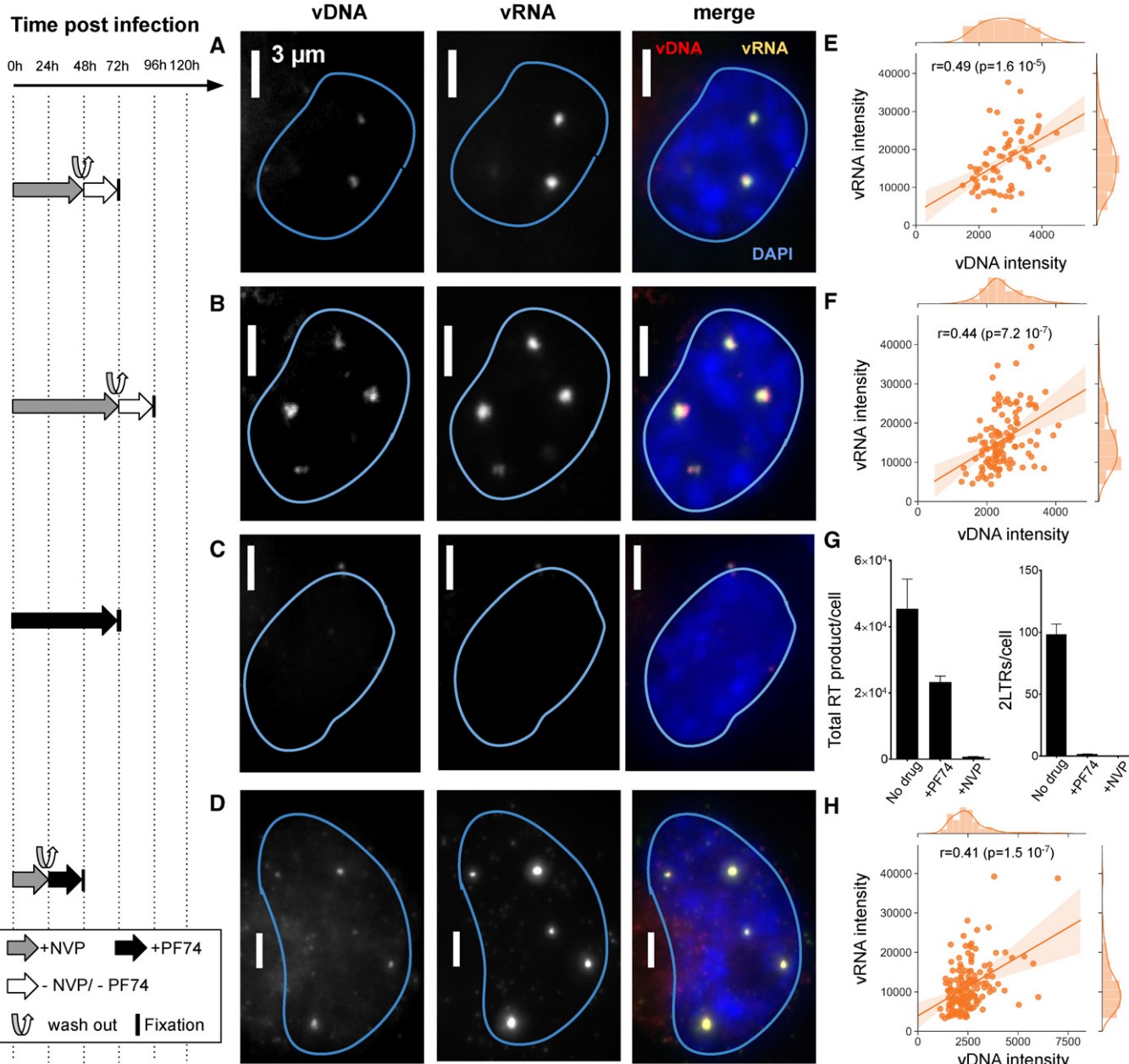

**Figure 5. Reverse transcription in nuclear clusters of infected macrophages.** The left panel shows the timeline of drug exposure experiments.

A–D   Images show vRNA and vDNA in an infected THP-1 cell (MOI 20) for each of four experimental conditions. See Appendix Figs S15–S17 for images of larger regions. (A, B) THP-1 cells were exposed to NVP for 48 h (A) or 72 h after infection (B), then NVP was washed out, and cells were fixed for click chemistry 24 h later. (C) Cells were exposed to PF74 for 72 h after infection, then fixed for click chemistry. (D) Cells were exposed to NVP for 24 h after infection, then NVP was washed out, and cells were exposed to PF74, before being fixed for click chemistry 24 h later.

E, F, H   Scatter plots show vDNA and vRNA intensities in colocalizing clusters with Spearman correlation *r* and associated *P*-values. Replicates of the four experiments yielded similar results.

G   qPCR measures DNA synthesis (left) and nuclear import (right) at 72 h p.i. in absence of drug treatment, or after exposure to NVP or PF74. Bars and error bars define mean and standard deviation, respectively. Samples were analyzed in triplicate and two biological replicates were performed.

experiments because we observed that EdU negatively interferes with viral transcription (Appendix Fig S21F). Overall, our data suggest that nuclear RT activity, at least in conditions where RT was initially suppressed, can lead to the production of transcription-competent viral DNA.

## Nuclear clustering and reverse transcription in primary macrophages

Finally, we asked if the main phenotypes described in THP-1 cells above also extend to primary macrophages. We therefore prepared

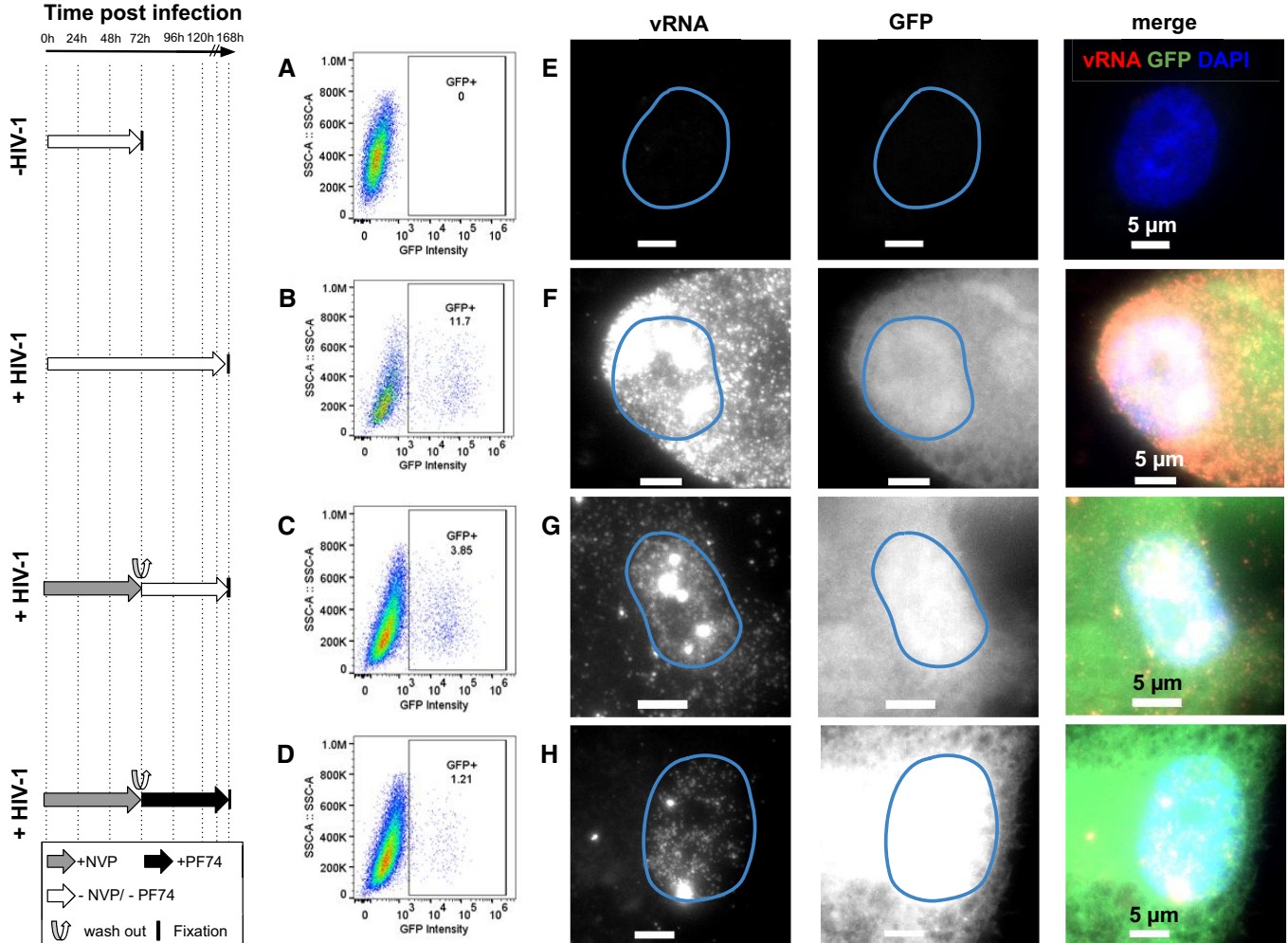

**Figure 6. Nuclear RT can yield transcription-competent vDNA.** The left panel shows the timeline of infection and/or drug exposure experiments.

A–D Percentage of GFP-positive cells at 3 and 7 days post-infection analyzed by FACS in absence of EdU. (A) Uninfected control cells. (B) Untreated, infected THP-1 cells fixed at 3 days (72 h) p.i. (C) Infected cells were exposed to NVP for 72 h, then NVP was washed out and cells were cultured for another 96 h and fixed at 7 days p.i. (D) Infected cells were cultured for 72 h p.i., then NVP was washed out and cells were exposed to the nuclear import inhibitor PF74 for another 96 h and fixed at 7 days p.i.

E–H Multicolor images of THP-1 cells infected with a GFP reporter virus. vRNA (RNA-FISH) is labeled in red, GFP in green, and nuclei (DAPI) in blue. (E) Uninfected control cells. (F) Untreated, infected THP-1 cells fixed at 3 days (72 h) p.i. (G) Infected cells were exposed to NVP for 72 h, then NVP was washed out and cells were cultured for another 96 h and fixed at 7 days p.i. (H) Infected cells were cultured for 72 h p.i., then NVP was washed out and cells were exposed to the nuclear import inhibitor PF74 for another 96 h and fixed at 7 days p.i. The experiment was repeated three times.

monocyte-derived macrophages (MDM) obtained from two healthy donors. FACS analysis indicated that ~ 76% of these cells were CD14 positive, confirming that the large majority consisted of macrophages (Appendix Fig S22). We then infected these MDMs with VSV-G-pseudotyped HIV-1ΔEnv. Cells from donor 1 were infected with the GFP reporter virus in presence of Vpx, while cells from donor 2 were infected with a virus without GFP and in absence of Vpx. We imaged the vDNA and vRNA using EdU and RNA-FISH, using similar experimental conditions as for the THP-1 cells above, including absence of treatment, treatment with NVP, and treatment with NVP followed by wash-out, (Figs 7 and EV4, Appendix Fig S23). In untreated MDMs at 7 d post-infection, we again observed bright nuclear foci of vDNA in a sizeable fraction of the cell population for both donors (15 out of 31 counted for

cells, i.e. ~ 50%, for cells, for donor 2 and 22 out of 53, i.e. ~ 30% for donor 1), thus both in presence and in absence of Vpx (Fig 7A and B, and Appendix Fig S23A and E). Focusing on cells from donor 2 (infected without Vpx), we measured vDNA foci sizes (FHWM) similar to those measured in THP-1 cells (median ~ 520 nm, interquartile range = 259 nm, $n = 20$ regions of interest) (Fig 7E). These vDNA foci also partially colocalize with vRNA foci (Fig 7A and B), as previously shown for THP-1 cells (Costes $P$-values < 0.05 in 18 out of $n = 20$ regions) (Fig 7F). Neither vDNA nor vRNA foci could be observed in uninfected MDM cells (Appendix Fig S23B and F).

MDM cells of donor 2 treated with 10 μM of NVP for 7 days also displayed vRNA foci, but not vDNA foci, indicating that genomic vRNA accumulates in nuclear foci of MDMs much as in THP-1 cells

(Fig 7C, Appendix Fig S23C). Moreover, in MDM cells treated with NVP for 3 days, followed by wash-out and fixation 4 days later, we again observed vDNA signal in vRNA clusters, consistent with a resumption of nuclear RT activity as previously observed for THP-1 cells (Fig 7D, Appendix Fig S23D). Similar observations were made for donor 1 (Fig EV4).

To determine if nuclear RT can lead to a transcriptionally competent vDNA template, we used FACS to analyze production of the HIV polyprotein Gag in MDMs from donor 2. Our data indicate that under the conditions used above to determine nuclear RT (NVP treatment for 3 days followed by wash-out and 4 days recovery), Gag median fluorescence intensity (MFI) is larger than in uninfected

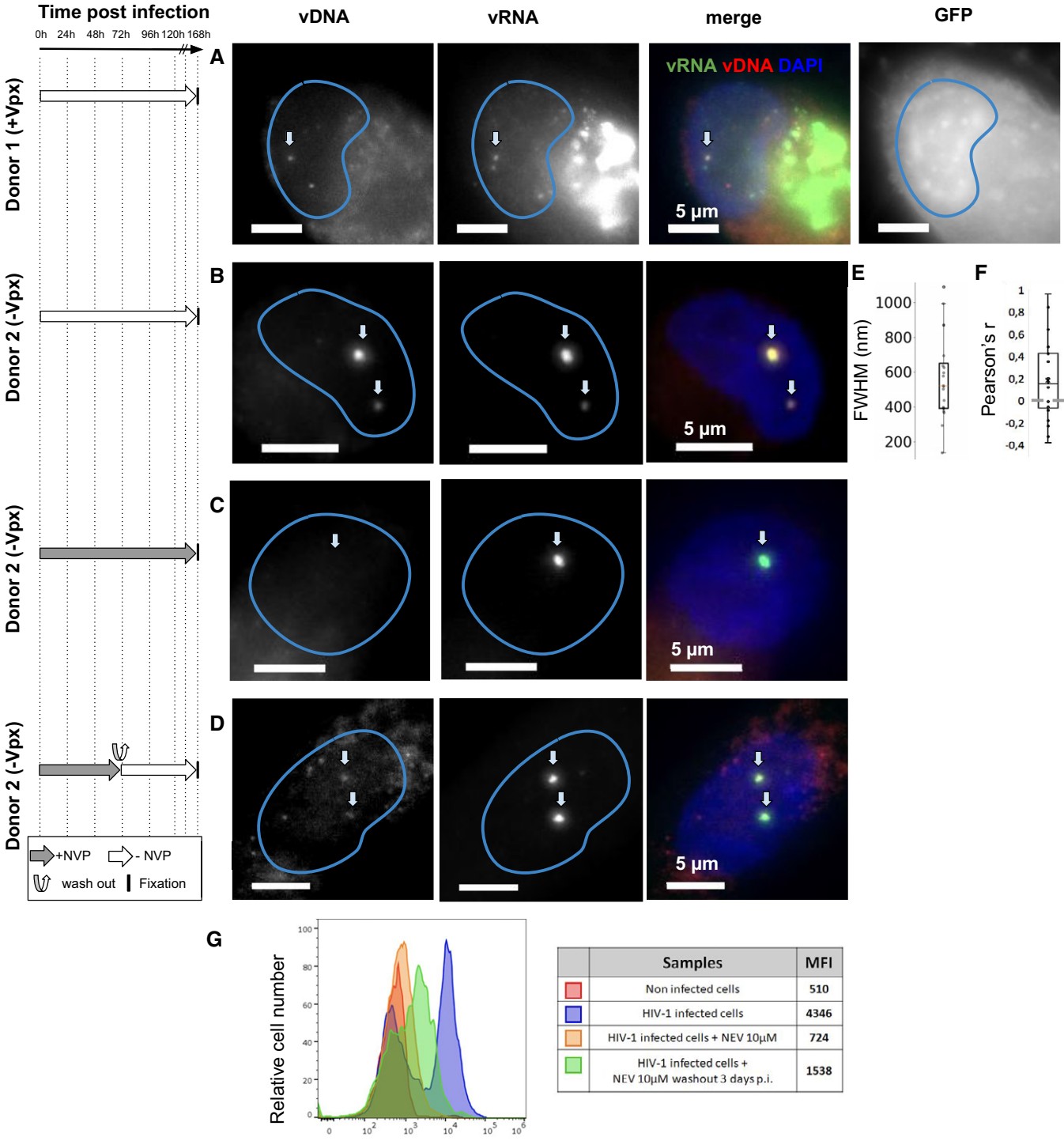

**Figure 7.**

**Figure 7.  Clustering of vDNA/vRNA and nuclear RT in primary human macrophages.**
The left panel shows the timeline of infection and/or drug exposure experiments.

A–D  Multicolor images of vDNA, vRNA, DAPI, and/or GFP in monocyte-derived macrophages (MDMs) from two different donors, infected with HIV-1. Cells from donor 1 were infected with VSV-G-pseudotyped HIV-1ΔEnv Vpx carrying a GFP reporter (A) and cells from donor 2 were infected with a VSV-G-pseudotyped HIV-1ΔEnv without Vpx (B–D). (A, B) Infected cells were left untreated and fixed at 6 d p.i. (C) Infected cells were treated with 10 μM Nevirapine (NVP) throughout the experiment and fixed at 7 days p.i. (D) Infected cells were exposed to NVP for 3 days p.i., then NVP was washed out and cells were cultured for another 4 days and fixed at 7 days p.i. Arrows in (A–D) show the position of selected vRNA foci.

E  Boxplot shows the size of DNA foci in untreated infected cells from donor 2 imaged at 7 days p.i. (B), measured by the FWHM of *n* = 24 foci. The red line defines the median, top, and bottom limits define upper and lower quartiles, respectively. Whiskers show the full data range, except for outliers. Gray dots are individual data points.

F  Boxplot shows Pearson's *r*, measuring colocalization of DNA with RNA foci in the same cells (*n* = 20). Center line defines the median, top, and bottom limits define upper and lower quartiles, respectively. Whiskers show the full data range, except for outliers. Black dots are individual data points.

G  FACS analysis of Gag positive cells from donor 2 in different conditions. Red: uninfected cells. Blue: untreated infected cells at 7 days p.i. as in (B). Orange: cells were treated with NVP for 7 days as in (C). Green: infected cells were exposed to NVP for 3 days, then NVP was washed out and cells were cultured for another 4 days as in (D). Median of fluorescence intensities (MFI) for each sample are shown in the table.

cells or cells treated with 10 μM NVP for 7 days (1,538 vs. 510 and 724, respectively) (Fig 7G). Under the same experimental conditions, we also observed some GFP-positive cells in MDMs from donor 1 (Fig EV4). These data support the notion that nuclear RT in MDMs can lead to synthesis of transcriptionally competent vDNA.

Thus, the main phenotypes detailed above in THP-1 cells can also be observed in MDMs, irrespective of the presence of Vpx. In summary, our data show that in primary macrophages, HIV-1 RNA genomes form RT-competent nuclear clusters that can lead to transcriptionally competent viral DNA.

# Discussion

Our study uses imaging of viral DNA and RNA to shed new light on the early replication cycle of HIV-1 in macrophage-like (THP-1) cells and in primary (monocyte-derived) macrophages. We demonstrated that vDNA and vRNA genomes cluster together in nuclear niches associated with speckle factors and that these clusters can form in absence of viral integration into the host genome. We further showed that genomic vRNA clusters can form in absence of vDNA after pharmacological inhibition of RT, but, importantly, also in untreated cells. Our observation of genomic RNA in nuclei agrees with previous studies showing that RT is dispensable for nuclear import (Burdick *et al*, 2013, 2017, 2020; Bejarano *et al*, 2019). However, the potential role of genomic vRNA in the nucleus remained unknown. Several recent studies have evoked the possibility of RT in the nucleus (Burdick *et al*, 2017, 2020; Bejarano *et al*, 2019). By combining reversible RT inhibition with direct visualization of the synthesized vDNA, our study provides the first clear demonstration that genomic RNA clusters can serve as templates for RT in the nucleus. We additionally showed that the viral DNA resulting from this nuclear RT activity can serve as a template for transcription. We note that two recent studies published during the revision of this paper reported similar findings (Francis *et al*, 2020; Dharan *et al*, 2020).

We emphasize that our data do not implicate that RT occurs exclusively, or even majoritarily, in the nucleus. Our results allow the possibility that RT is initiated in the cytoplasm and prolonged in the nuclear compartment, or that some viral RNA genomes are entirely reverse transcribed in the cytoplasm. We also acknowledge that our demonstration of nuclear RT was achieved after temporary pharmacological inhibition of RT. Nevertheless, our observation

of nuclear genomic vRNA clusters in absence of drug treatment opens the possibility that RT also occurs in the nuclei of untreated macrophages.

Our clear evidence for nuclear RT stands in stark contrast with the classical picture of the HIV replication cycle, according to which RT is entirely restricted to the cytoplasmic compartment. An intriguing speculation is that these vRNA clusters may act as nuclear microreactors that concentrate viral reverse transcriptase enzymes to enable efficient vDNA synthesis in macrophages, much as intact capsid cores are believed to concentrate these enzymes in the cytoplasm of HeLa or CD4$^+$ cells (Warrilow *et al*, 2007).

Previous *in vitro* experiments have suggested a potential link between HIV-1 and nuclear speckles (Bell *et al*, 2001; Pendergrast *et al*, 2002), and earlier studies have reported that HIV-1 RNA colocalizes with the speckle factor SC35 (Bøe *et al*, 1998; Cardinale *et al*, 2007). Here, we directly show that HIV-1 RNA/DNA genome clusters localize to nuclear niches enriched in SC35. While it has been proposed that the nuclear pool of unspliced HIV-1 RNAs may be stored in nuclear paraspeckles (Zhang *et al*, 2013), our data instead indicate that viral clusters sequester the host cell factor CPSF6 away from its canonical localization in paraspeckles (Bøe *et al*, 1998; Cardinale *et al*, 2007; Burdick *et al*, 2020) and relocate this protein in speckles. Our data are consistent with a recent study demonstrating the role of CPSF6-CA interactions in mediating the association of HIV-1 with nuclear speckles (Francis *et al*, 2020).

Several reports highlighted the ability of other viruses to alter the cytoplasmic or nuclear organization and form microenvironments that locally concentrate viral and host cell factors required for the synthesis of viral progeny and/or to shield the virus from cellular defense mechanisms (Schmid *et al*, 2014; Heinz *et al*, 2018; McSwiggen *et al*, 2019). An interesting possibility is that the crowded microcompartments formed by cellular RNAs in these nuclear bodies may shroud the viral DNA and protect it from mediators of the innate immune response, such as the cyclic GMP-AMP synthase (cGAS), a viral DNA sensor that associates with HIV DNA in nuclei of infected macrophages (Lahaye *et al*, 2018; Sumner *et al*, 2020).

Our observation of vRNA/vDNA clusters in cells infected by an integration-defective virus or upon inhibition of integration further suggests that most of the vDNA in nuclear clusters may be unintegrated. This dovetails with the common observation of large amounts of unintegrated viral DNA in the nucleus, which can act as viral reservoirs and constitute an important obstacle to successful treatment of HIV-1 infection (Bell *et al*, 2001; Gelderblom *et al*,

2008; Hamid *et al*, 2017). Thus, the unintegrated vDNA observed in our study may be relevant to understanding HIV reactivation in patients.

More research is needed to further explore the functional role and formation mechanisms of these vRNA/vDNA clusters. Our discovery of nuclear RT-competent clusters of viral RNA and DNA in THP-1 cells and primary macrophages opens new perspectives for understanding and hence combating HIV-1 replication in natural target cells.

# Materials and Methods

## Cell culture

Human THP-1 cells (ATCC TIB-202) were grown in RPMI 1640 medium supplemented with 10 % (vol/vol) fetal bovine serum and 1% (vol/vol) penicillin-streptomycin. For infections, 12-well plates containing coverslips were seeded with $1 \times 10^6$ THP-1 cells and treated with PMA (259 µg/ml final concentration) at 37°C and 5% $CO_2$. Twenty four hours post-stimulation, non-adherent cells were removed, adherent cells were washed, and further cultured. PMA was present during the entire experiment.

## Plasmids and viral production

The plasmid HIV-1ΔEnv IN$_{HA}$ (D116A)ΔNef was obtained by insertional mutagenesis using the QuikChange II XL Site-Directed Mutagenesis kit (Agilent). HIV-1 viruses were produced by cotransfection with calcium phosphate with 10 µg HIV-1 LAI (BRU) (or NL4.3) ΔEnv Virus (NIH) or with the modified versions HIV-1ΔEnvIN$_{HA}$ (Petit *et al*, 1999) or HIV-1ΔEnv IN$_{HA}$ (D116A)ΔNef in combination with 1 µg of VSV-G envelope expression plasmid pHCMV-G (VSV-G) with or without 3 µg of SIV$_{MAC}$ Vpx (Durand *et al*, 2013). The viruses collected from 293T cells 48 h post-transfection were ultracentrifuged at 4 °C for 1h at 87,275 × *g* at r-max. Virus normalizations were performed by p24 ELISA according to the manufacturer's instructions (Perkin Elmer) or by qPCR. HIV-1 ΔEnv Δnef LUC and HIV-1 ΔEnv Δnef GFP viruses were produced and titered similarly.

## Quantitative PCR

DNA synthesis, nuclear import, and integration during HIV-1 infection in THP-1 cells were quantified by qPCR. We analyzed for late reverse transcription (LRT) products representing HIV-1 DNA synthesis in the cell and 2LTRs to measure nuclear import by qPCR. Integration of proviruses into the human genome was measured by ALU PCR. Viruses were treated for 30 min at 37°C with 1,000 U of DNase I (Roche). As a control, 10 µM nevirapine was used in infected cells. Total cellular DNA was isolated using the QIAamp DNA micro kit (Qiagen) at 7 and 24 h p.i. Viral DNA synthesis products and 2LTRs were measured at different time points by real-time PCR. LRT, 2LTRs, and Alu PCR were performed as described (Di Nunzio *et al*, 2013; Lelek *et al*, 2015). LRT was amplified using the primers MH531 and MH532, with the standard curve prepared using the plasmid coding for the viral genome. 2LTRs were amplified by primers MH535/536 and probe MH603, using as standard curve the pUC2LTR plasmid which contains the HIV-1 2LTR junction.

Integration was assessed by Alu-PCR, using primers designed in the U3 region of LTR (Di Nunzio *et al*, 2013; Lelek *et al*, 2015). The standard curve was prepared as follows: DNA generated from infected cells was end point diluted in DNA prepared from uninfected cells and serial dilutions were made. The control of the first-round PCR was the amplification without Alu primers but only U3 primers (Di Nunzio *et al*, 2013; Lelek *et al*, 2015). Dilutions of the first round were processed by real-time PCR (Di Nunzio *et al*, 2013; Lelek *et al*, 2015). All experiments were carried out using internal controls such as infection in presence of RAL (10 µM) and/or NVP (10 µM). LRT, 2-LTR, and Alu-PCR reactions were normalized by amplification of the Actin gene (Di Nunzio *et al*, 2013; Lelek *et al*, 2015).

## Virus infection

Cells were infected at least 24 h after initial PMA stimulation with MOIs 10, 50, or 100, based on the viral titer calculated on 293T cells ($8.93 \times 10^8$ TU/ml) by qPCR. For experiments with antiretroviral drugs, cells were infected with HIV-1 in the presence of 10 µM RAL or 10 µM NVP or 1.5 µM PF74.

## Drug wash-out experiment

Cells were grown and differentiated as described above. Cells were infected with an MOI of 50 in the presence of EdU and NVP or PF74. For experiments where NVP was washed out, the drug-containing medium was removed, cells were washed twice with 1 ml of medium for 15 min, fresh medium was added and cells were incubated for the indicated amount of time. The medium was changed every 24 h and reconstituted with or without the PF74 drug, as indicated.

## Labeling of viral DNA by click chemistry and immunolabeling

For imaging of vDNA, CA, and IN, THP-1 cells were seeded in 12-well plates on cover glass in the presence of PMA and infected on the following day with HIV-1 at different MOI in medium containing 5 µM EdU. After 24 h, the medium was removed and replaced by fresh medium containing 5 µM EdU and incubation was continued at 37°C. To stop the infection, cells were washed with warm PBS and fixed with 4% paraformaldehyde in PBS for 20 min at room temperature.

For vDNA labeling, cells were washed twice with PBS supplemented with 3% bovine serum albumin (BSA) and permeabilized with 0.5% (vol/vol) Triton X-100 for 30 min. After washing with 3% BSA in PBS twice, click-labeling was performed for 30 min at room temperature using the Click-iT EdU- Alexa Fluor 647 Imaging Kit (Thermo Fisher Scientific) following the manufacturer's instructions.

For immunolabeling, cells were blocked for 30 min with 3% BSA in PBS and permeabilized with 0.5% (vol/vol) Triton X-100 for 30 min. After two washes with 3% BSA in PBS, cells were incubated with the primary antibody in 1% BSA in PBS for 1 h at room temperature. After washing with 1% BSA in PBS, a secondary antibody was used for 1 h at room temperature in 1% BSA in PBS. The primary and secondary antibodies used in this study are listed in Appendix Note S1.

In experiments that combine EdU and protein labeling, the click chemistry reaction was performed prior to immunolabeling, and blocking with 3% BSA was omitted.

## Labeling of viral RNA with single-molecule RNA-FISH

To visualize individual vRNA molecules, we used the smiFISH approach (Tsanov *et al*, 2016). Unlabeled primary probes are designed to target the RNA of interest and can be pre-hybridized with fluorescently labeled secondary detector oligonucleotides for visualization. All probes, except against Neat1, were designed using either OLIGOSTAN (Tsanov *et al*, 2016) or Stellaris Probe designer, and purchased from Integrated DNA Technologies (IDT). All probe sequences are available in Appendix Tables S1–S3. We used 24 primary probes, each 18–20 nt long, against the HIV-1 POL gene. Probe sets against GFP and LUC comprised 18 and 22 probes, respectively. Secondary probes are conjugated to either Cy3 or Cy5. To detect Neat1, we used Stellaris FISH probes against human NEAT1 5′ segment with QUASAR 570 DYE SMF-2036-1.

Cells were fixed as described above, washed twice with PBS, and stored in nuclease-free 70% ethanol at −20°C until labeling. On the day of the labeling, the samples were brought to room temperature, washed twice with wash buffer A (2× SSC in nuclease-free water) for 5 min, followed by two washing steps with washing buffer B (2× SSC and 10% formamide in nuclease-free water) for 5 min.

The target-specific primary probes were pre-hybridized with the fluorescently labeled secondary probes via a complementary binding readout sequence. The stock concentration of the probes was as follows: Luciferase LUC: 148.6 ng/μl, POL: 1,497 ng/μl, and GFP: 163.3 ng/μl.

The reaction mixture contained primary probes at a final concentration of 40 pm and secondary probes at a final concentration of 50 pm in 1× NEB3 buffer. Pre-hybridization was performed in a PCR machine with the following cycles: 85°C for 3 min, followed by heating to 65°C for 3 min, and a further 5 min heating at 25°C. 2 μl of this FISH-probe stock solution was added to 100 μl of hybridization buffer (10% (*w/v*) dextran, 10% formamide, 2× SSC in nuclease-free water).

Samples were placed on Parafilm in a humidified chamber on 100 μl of hybridization solution, sealed with Parafilm, and incubated overnight at 37°C. The next day, cells were washed in the dark at 37°C without shaking for > 30 min twice with pre-warmed washing buffer B. Sample were washed once with PBS for 5 min, stained with DAPI in PBS (1:10,000) for 5 min, and washed again in PBS for 5 min. Samples were mounted in ProLong Gold antifade mounting medium.

For simultaneous detection of viral DNA with click chemistry and vRNA with RNA-FISH, the click reaction was performed prior to RNA-FISH and as described above, except no BSA was used in order to minimize RNA degradation by RNases. The ratio copper-protectant/copper solution was 1.5:1 to avoid degradation of RNA, as indicated by the manufacturer.

## Monocyte isolation, differentiation, and infection

Peripheral blood mononuclear cells (PBMCs) were isolated from two HIV-seronegative donors (Etablissement français du sang, EFS), by density-gradient centrifugation. Monocyte-derived macrophages (MDMs) were prepared by adherence with washing of non-adherent cells after 2 h. Adherent cells were selected in RPMI 1640 medium supplemented with 10% human serum or 10% fetal calf serum (FCS) and MCSF (10 ng/ml) for 3 days and then differentiated for another 4 days in RPMI 1640 medium supplemented with 10% FCS without MCSF. FCS was used to prepare stimulated cells following the protocol described in Mlcochova et al. (Mlcochova *et al*, 2017). MDMs of donor 1 were infected with MOI 100 of HIV-1 GFP in the presence of Vpx. MDMs of donor 2 were infected with 500 ng of p24 of the HIV-1 without Vpx. Samples in which RT was inhibited, 10 μM NVP was present for indicated periods of time. Samples used for click chemistry staining contained 5 μM final concentration of EdU. MDMs were fixed in PFA 4% for 20 min at RT, then washed twice with PBS. Samples were frozen at −20° in 70% cold ethanol for at least 12 h before click chemistry and RNA-FISH (see sections above). Duplicates of samples without EdU were analyzed by FACS and stained in permeabilized samples with 0.05% Saponin using anti-Gag antibody conjugated to Rhodamine (Beckman Coulter, #6604667) diluted 1:500 in PBS-BSA 0.5%. MDMs were controlled for the expression of CD14 (Ab anti-CD14-FITC,BD #555397, dilution 1:20) in unfixed cells.

## Polymer simulations

To estimate the expected size and images of single vDNA particles, we performed polymer simulations in which the vDNA molecule is represented by a semiflexible chain of beads undergoing random motions (Langevin dynamics) (Arbona *et al*, 2017). We assumed chromatinized (i.e., nucleosome-containing) DNA and performed two types of simulations: one with a linear chain (as expected e.g., for integrated vDNA) and the other with a circular chain (e.g., for unintegrated and episomal vDNA). Each bead represents a single nucleosome consisting of 175 bp of DNA (nucleosomal + linker DNA) confined in a 11 nm spherical bead. We assumed the bending stiffness of the polymer to be 11 nm (single bead) and connected consecutive beads via spring potentials. We started our simulations from a linear or circular configuration and allowed the polymer to reach equilibrium (as judged by the temporal evolution of the gyration radius) before sampling one configuration for each simulation. Simulations were run using the LAMMPS simulation library (https://lammps.sandia.gov/). Simulated images were obtained by convolving the simulated configuration with a 3D Gaussian kernel of standard deviation 100 nm approximating the microscope point spread function. We used 100 independent simulations to compute the distributions of sizes (FWHM) for each model.

## Imaging and analysis

Three-dimensional image stacks with a z-spacing of 200 nm were captured on a wide-field microscope (Nikon TiE Eclipse) equipped with a 60× 1.4 NA objective and an sCMOS camera (Hamamatsu Orcaflash 4), with a lumencor SOLA LED light source with adequate Semrock filters, and controlled with MicroManager 1.4.

Nuclei were automatically detected by a trained neural network implemented within the computational platform ImJoy (Ouyang *et al*, 2019) (Plugin DPNUnet). Spots in Edu and FISH images were detected with a standard spot detection approach: images were first filtered with a Laplacian of Gaussian filter (LoG), then spots were identified with a local maximum filter with a user-defined minimum intensity.

Colocalization analysis between two channels was implemented in a custom ImJoy plugin. Only spots located within the detected

nuclei were considered. Colocalization was implemented as a linear assignment problem (Python function linear_sum_assignment from SciPy), where each spot from the first channel is assigned to one spot in the second channel. Each spot can be only assigned once, and only assignments below a user-defined distance threshold are permitted. Spot intensities for colocalized spots were measured as the maximum intensity in a $\pm$ 1 pixel window. To calculate *P*-values for the reported colocalization, the colocalization analysis was repeated 100 times after randomly moving ("jitter") detected spots in one channel by 1,000 nm in XYZ. The reported *P*-values are the proportion of jitter iterations where a higher colocalization was obtained than in absence of jitter.

EdU enrichment for Nevirapine experiments was calculated after nuclei segmentation and spot detection as described above. Only RNA foci within the detected nuclei and with intensities above a user-defined threshold were considered. FISH and EdU intensities at these locations were measured as the maximum intensity in a window of $\pm$ 300 nm.

To analyze colocalization of immunolabeled proteins and Neat1 with vDNA, we used the Costes method (Costes *et al*, 2004) as implemented in the coloc2 plugin of Fiji (Schindelin *et al*, 2012). First, nuclei were manually selected to exclude background that could lead to inappropriate thresholding during correlation analysis. For each selected region, the plugin automatically thresholds each channel such that the Pearson correlation coefficient of channel intensities below the threshold is 0. The Pearson correlation of intensities above the thresholds is then reported. Similarly to above, statistical significance was determined by comparing this correlation to that of 100 randomly jittered images, where jittering consisted in moving one channel by 3 pixels. If the Pearson correlation of the unjittered channels is positive (respectively, negative), the reported *P*-value reflects the proportion of jitter iterations in which the Pearson coefficient is higher (respectively, lower).

To estimate the size of nuclear vDNA foci, we used Fiji (Schindelin *et al*, 2012) to manually draw line profiles across EdU spots. Each intensity profile was then fitted with a Gaussian function and the corresponding full width at half maximum (FWHM) was used as a measure of the vDNA cluster size.

## Data availability

This study includes no data deposited in external repositories.

**Expanded View** for this article is available online.

## Acknowledgements
We thank Mickaël Lelek and Andrey Aristov for help with microscopy and Benoît Lelandais for help with image analysis. We thank Marie-Anne Welti, Edouard Bertrand, Xavier Darzacq, Olivier Schwartz, Arnaud Echard, and Michaela Mueller for useful discussions and/or comments on the manuscript. We thank Fabrizio Mammano for kindly providing the modified IN-HA HIV-1 virus and Stella Frabetti for technical support. We also thank the UtechS Photonic BioImaging, C2RT, Institut Pasteur, which is supported in part by the French National Research Agency (France BioImaging; ANR-10–INSB–04; Investments for the Future). We thank the NIH AIDS Reagents program for reagents. We also acknowledge Investissement d'Avenir grant ANR- 16-CONV-0005 for funding computing resources used for simulations. ER was funded by a Fondation pour la Recherche Médicale en France (FRM) grant to CZ (DEQ 20150331762), an Agence Nationale de la Recherche sur le Sida et les hépatites virales fellowship to ER (ANRS, ECTZ74440) associated with an ANRS grant to FDN (ECTZ4469), and a fellowship by Institut Carnot Pasteur MS to ER associated with CZ laboratory. We acknowledge additional funding by Institut Pasteur, FRM/Sidaction (grant VIH20170718001 to FM and FDN), and ANRS (grant ECTZ88162 to FDN).

## Author contributions
ER, FM, CZ, and FDN conceived and designed experiments. ER performed all imaging. ER, VS, and FDN performed THP-1 experiments. VS, PS, and FDN performed qPCR, FACS, and primary macrophages experiments. ER and FM analyzed imaging data. JJP performed simulations. CZ and FDN obtained grants. ER obtained fellowships. FM, CZ, and FDN supervised project. Wrote paper: CZ, with major input from ER, FM, and FDN.

## Conflict of interest
The authors declare that they have no conflict of interest.

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
