## [Review Process File · The EMBO Journal]

Clustering and reverse transcription of HIV-1 genomes in nuclear niches of macrophages

Elena Rensen, Florian Mueller, Viviana Scoca, Jyotsana Parmar, Philippe Souque, Christophe Zimmer, and Francesca Di Nunzio

DOI: [10.15252/embj.2020105247](https://doi.org/10.15252/embj.2020105247)

Corresponding author(s): *Christophe Zimmer (czimmer@pasteur.fr)*, *Florian Mueller (florian.muller@pasteur.fr)*, *Francesca Di Nunzio (francesca.di-nunzio@pasteur.fr)*

Review Timeline:

Submission Date:	9th Apr 20
Editorial Decision:	28th May 20
Revision Received:	24th Aug 20
Editorial Decision:	18th Sep 20
Revision Received:	4th Oct 20
Accepted:	16th Oct 20

Editor: Karin Dumstrei

Transaction Report:

Dear Christophe

Thank you for submitting your manuscript to The EMBO Journal. Your study has now been seen by two referees and their comments are provided below.

Both referees appreciate the reported findings and I would like to invite you to submit a revised manuscript. The referees indicate that some further analysis in primary MDM macrophages would strengthen the findings. I realise that referee #2 suggests that given the current circumstance that probably OK to forgo such experiments, but I would very much like to see if it would be possible to carry out such an experiment. I think it would be helpful to discuss the raised points further via phone or skype.

When preparing your letter of response to the referees' comments, please bear in mind that this will form part of the Review Process File, and will therefore be available online to the community. For more details on our Transparent Editorial Process, please visit our website:

<https://www.embopress.org/page/journal/14602075/authorguide#transparentprocess>

Thank you for the opportunity to consider your work for publication. I look forward to discussing the revisions further with you

best Karin

Karin Dumstrei, PhD
Senior Editor
The EMBO Journal

- a point-by-point response to the referees' comments, with a detailed description of the changes made (as a word file).
- a word file of the manuscript text.
- individual production quality figure files (one file per figure)
- a complete author checklist, which you can download from our author guidelines (<https://www.embopress.org/page/journal/14602075/authorguide>).

- Expanded View files (replacing Supplementary Information)

Further information is available in our Guide For Authors:

The revision must be submitted online within 90 days; please click on the link below to submit the revision online before 26th Aug 2020.

Referee #1:

The authors present a study on nuclear events in the HIV life cycle in myeloid cells. The HIV field has until recently viewed RT as occurring in the cytoplasm and more recently at the nuclear pore and inside the nucleus. Here the authors suggest RT can occur in the nucleus of THP1 cells using co-infection with HIV and a Vpx expressing SIV. The topic is interesting to read though I have a number of concerns:

1. All the work presented is done in PMA treated THP-1 cells. Authors should show that nuclear viral DNA and RNA foci can be detected in primary MDM macrophages in order to justify that this is indeed a macrophage phenomenon. In order to increase the dNTP and RT levels and make vDNA accumulation detectable, the authors could culture MDM in FCS as per Mlcochova et al.2017.
2. Vpx as a system for knocking down SAMHD1 is problematic due to its known effect on HUSH (PMID 30297740, 29891865) and possible other effects on the cell. By performing the experiment in 1. The authors would not need to use SIV expressing Vpx as SAMHD1 would be inactive under those culture conditions. There is the possibility that the SIV has an impact on the observations and we would need to see controls for a Vpx deleted SIV. They could additionally use siRNA knockdown of SAMHD1 in MDM or in THP-1.
3. The foci appear to be not a feature of productive infection as the authors point out with the integrase experiment. With 73% of cells having RT foci and only 1.8% infection, the most likely interpretation is that we are observing largely abortive events. Could the authors show that when the MOI is reduced that the infection % is preserved to some extent? The infection data are key to interpretation and should be in the main figures rather than supplementary.

Minor

Other groups have suggested that capsid uncoating occurs just before integration and that RT occurs in the nucleus. The authors show capsid colocalisation with vRNA and DNA clusters but did not explore the uncoating event here. A clear discussion on uncoating and the role of CPSF6 in nuclear import was missing from the paper and is of importance.

Referee #2:

In this paper, Rensen et al report that, in differentiated THP-1 cells, which are non-dividing and monocytic in character, the HIV-1 vRNA and viral DNA form intranuclear structures that can perform at least some level of reverse transcription. The imaging data are nice but largely phenomenological as it is unclear why these structures form or what their purpose is. The paper would also have been greatly strengthened by at least some experiments performed in primary monocyte-derived macrophages. Nevertheless, the data are clearly surprising and interesting and the paper is well written and balanced. Given the current circumstances, my inclination is to say that, despite minor caveats, this research is worth bringing to the attention of the field.

We thank the reviewers for their constructive comments on our manuscript, which led us to perform important new experiments, as detailed below.

Referee #1:

The authors present a study on nuclear events in the HIV life cycle in myeloid cells. The HIV field has until recently viewed RT as occurring in the cytoplasm and more recently at the nuclear pore and inside the nucleus. Here the authors suggest RT can occur in the nucleus of ThP1 cells using co-infection with HIV and a Vpx expressing SIV. The topic is interesting to read though I have a number of concerns:

1. All the work presented is done in PMA treated THP1 cells. Authors should show that nuclear viral DNA and RNA foci can be detected in primary MDM macrophages in order to justify that this is indeed a macrophage phenomenon. In order to increase the dNTP and RT levels and make vDNA accumulation detectable, the authors could culture MDM in FCS as per Mlcochova et al.2017.

We agree that repeating the experiments in primary macrophages is important. Despite delays due to the Covid situation, we obtained monocyte derived macrophages (MDMs) from two human donors. We cultured these MDM cells in FCS following the protocol recommended by the reviewer, infected cells from donor 1 with a GFP-labelled virus in presence of Vpx, and donor 2 with a virus (without GFP) in absence of Vpx. We then imaged viral DNA and RNA using EdU and RNA-FISH as for ThP1 cells.

The results are reported in the new **Figure 7** and associated **Figures S23** and **Expanded View (EV) 4**.

We again observed bright colocalizing nuclear foci of vDNA and vRNA in cells 144 h post infection, both for donor 1 and for donor 2 (**Figure 7A,B** and **Figure S23A**). This confirms that the vDNA/vRNA cluster phenotype first revealed in ThP1 cells is indeed a macrophage phenomenon, addressing the referee's request.

Furthermore, we treated the MDMs with Nevirapine, and again observed vRNA foci (but no vDNA foci), confirming that the vRNA foci consist at least in part of genomic vRNA (**Figure 7C, S23C**), as we had shown for ThP1 cells.

In addition, we performed a Nevirapine wash-out experiment, and again observed the appearance of vDNA in vRNA foci, also indicating that the vRNA in these foci can undergo nuclear RT (**Figure 7D, S23D**).

Finally, under the same Nevirapine wash-out condition, FACS analysis indicates production of Gag (**Figure 7G**), and some cells infected with a GFP reporter virus are GFP positive (**EV 4**), suggesting that nuclear RT can lead to transcriptionally competent viral DNA.

Thus, our new data show that the main phenotypes described in ThP1 cells also hold true for primary macrophages. We believe that these additional data strongly underscore the relevance of our study for understanding HIV replication in patients.

The new data are now described in a new section at the end of the Results section.

2. Vpx as a system for knocking down SAMHD1 is problematic due to its known effect on HUSH (PMID 30297740, 29891865) and possible other effects on the cell. By performing the experiment in 1. The authors would not need to use SIV expressing Vpx as SAMHD1 would be inactive under those culture conditions. There is the possibility that the SIV has an impact on the observations and we would need to see controls for a Vpx deleted SIV. They could additionally use siRNA knockdown of SAMHD1 in MDM or in THP-1.

We acknowledge that the effect of Vpx on HUSH is a potential concern. However, the new experiments on primary macrophages were done both with and without Vpx and led to the same clustering of vDNA and vRNA in nuclear foci (**Figure 7A,B, S23 and EV4**). This shows that the observed phenotype is not an artifact due to the presence of Vpx.

3. The foci appear to be not a feature of productive infection as the authors point out with the integrase experiment. With 73% of cells having RT foci and only 1.8% infection, the most likely interpretation is that we are observing largely abortive events. Could the authors show that when the MOI is reduced that the infection % is preserved to some extent? The infection data are key to interpretation and should be in the main figures rather than supplementary.

We acknowledge that the 1.8% of GFP positive cells measured in our previous experiment (**Fig S21E**) suggested that only a small - though still significant- proportion of the cells was capable of transcription. However in the meantime we found out that this low number can be attributed at least in part to an inhibitory effect of EdU on transcription. Indeed, our new FACS analysis, now included as **Figure S21F**, shows that 7 days post infection the percentage of GFP positive cells is ~2% with EdU staining (similar to the 1.8% measured by microscopy), but ~11% without EdU staining.

Please note that this effect of EdU on transcription does not alter any of our other findings (clustering of viral genomes, presence of vRNA in the nucleus, nuclear RT) besides those on transcription.

We have now revisited the analysis of transcription using FACS in absence of EdU labeling (new **Figure 6A-D** for Thp1 cells, **Figure S21G-H** for MDMs). Our new FACS data indicates that in conditions where RT is confined to the nucleus (NVP treatment for 3 days followed by wash-out and treatment with PF74 to impeded nuclear import and fixation 4 days later), we can still detect ~1% of GFP positive cells at 7 d p.i. While this is much lower than the ~11% measured for untreated cells, it nevertheless indicates that some vDNA synthesized in the nucleus is in fact transcribed and then translated.

As suggested by the reviewer, we have now dedicated a main Figure (new **Figure 6A-D**) and Results section to these data.

Minor

Other groups have suggested that capsid uncoating occurs just before integration and that RT occurs in the nucleus. The authors show capsid colocalisation with vRNA and DNA clusters but did not explore the uncoating event here. A clear discussion on uncoating and the role of CPSF6 in nuclear import was missing from the paper and is of importance.

We only used capsid staining in **Fig. 1D** as a means to validate EdU labeling (along with IN) and did not analyze capsid localization thereafter. The resolution of these images does not allow us to draw any conclusion as to whether the capsid signal in Fig. 1D reflects the presence of intact capsid cores (as suggested by some) or multiple CA proteins or monomeric capsid that remains associated with the viral genome. Thus, our study does not directly bear on capsid uncoating. We feel that a discussion of capsid uncoating here would be distracting from the main focus of our paper.

However, we modified our Discussion to account for two recent papers from the Campbell and Melikyan groups that report similar findings concerning nuclear reverse transcription and/or association of HIV to speckles and the role of capsid-CPSF6 interaction.

Referee #2:

In this paper, Rensen et al report that, in differentiated THP-1 cells, which are non-dividing and monocytic in character, the HIV-1 vRNA and viral DNA form intranuclear structures that can perform at least some level of reverse transcription. The imaging data are nice but largely phenomenological as it is unclear why these structures form or what their purpose is. The paper would also have been greatly strengthened by at least some experiments performed in primary monocyte-derived macrophages. Nevertheless, the data are clearly surprising and interesting and the paper is well written and balanced. Given the current circumstances, my inclination is to say that, despite minor caveats, this research is worth bringing to the attention of the field.

We thank the reviewer for appreciating the interest of our data. We acknowledge that our study does not address the mechanism of formation of these intranuclear structures. We also did not fully address their purpose, although our observation of transcription competent nuclear RT suggests that these structures may be important for viral replication.

Our paper clearly establishes the existence of genomic vRNA and vDNA clusters and the possibility of reverse transcription in these clusters. Future work is needed to address the mechanisms and functional consequences of these surprising structures. We provide some hypotheses in the Discussion.

We agree with the reviewer about the importance of verifying our results in primary macrophages, as also pointed out by referee #1. Please see our detailed response above.

Dear Christophe,

Thanks for sending us your revised manuscript. I have now looked at and I appreciate the added data using primary macrophages. It really strengthens the analysis. The revised version has also been seen by referee #2 who as you can see below also appreciates the introduced revisions.

I am therefore very pleased to let you know that we will accept the manuscript for publication here. Before sending you the formal acceptance letter you just have to sort out the below listed editorial points. You can upload the revised version using the link below.

- We need 3-5 keywords
- We require a Data Availability section. As far as I can see no data is provided in the study that needs to be deposited in external database if so please add: This study includes no data deposited in external repositories
- we also need an acknowledgement section
- You have listed both corresponding and co-senior authors on the title page. Just use corresponding authors
- Please make sure to add funding info to the submission system (ECTZ4469, ECTZ74440 + ECTZ88162)
- Double check that there are figure calls out to Fig 3E and Fig 5D as well as to EV figure panels
- Take a look at Fig1- panel I is missing
- Please check if scale bars are needed in Figure EV1 A & B
- The Appendix figures and tables are missing the word 'Appendix' - see also guide to authors. Please correct callouts in manuscript text.
- The list of authors in the reference list should be cut after 10 followed by et al.
- The legend to the movie should be zipped with is legend. Please also correct nomenclature - should be Movie EV1
- We include a synopsis of the paper (see <http://emboj.embopress.org/>). Please provide me with a general summary statement and 3-5 bullet points that capture the key findings of the paper.
- We also need a summary figure for the synopsis. The size should be 550 wide by [200-400] high (pixels). You can also use something from the figures if that is easier.
- I have asked our publisher to do their pre-publication checks on the paper. They will send me the file within the next few days. Please wait to upload the revised version until you have received their comments.

That should be all! Congratulations on a nice study

With best wishes

Karin

Karin Dumstrei, PhD
Senior Editor
The EMBO Journal

Further information is available in our Guide For Authors:

The revision must be submitted online within 90 days; please click on the link below to submit the revision online before 17th Dec 2020.

Referee #2:

This interesting paper is much improved by the new data in primary macrophages in figure 7 and is now, in my view, appropriate for publication in EMBO journal. I have no further criticisms.

Point-by-point response, Oct 4, 2020

Dear Christophe,

Thanks for sending us your revised manuscript. I have now looked at and I appreciate the added data using primary macrophages. It really strengthens the analysis. The revised version has also been seen by referee #2 who as you can see below also appreciates the introduced revisions.

I am therefore very pleased to let you know that we will accept the manuscript for publication here. Before sending you the formal acceptance letter you just have to sort out the below listed editorial points. You can upload the revised version using the link below.

- We need 3-5 keywords

We added 5 keywords to the title page

- We require a Data Availability section. As far as I can see no data is provided in the study that needs to be deposited in external database if so please add: This study includes no data deposited in external repositories

We added this statement at the end of the Methods section (after the Methods references)

- we also need an acknowledgement section

We have an acknowledgement section after the Discussion

- You have listed both corresponding and co-senior authors on the title page. Just use corresponding authors

We did so, but indicated equal contributions for the co-senior authors

- Please make sure to add funding info to the submission system (ECTZ4469, ECTZ74440 + ECTZ88162)

Done

- Double check that there are figure calls out to Fig 3E and Fig 5D as well as to EV figure panels

We added the missing figure call outs

- Take a look at Fig1- panel I is missing

We relabeled the panel J as panel I

- Please check if scale bars are needed in Figure EV1 A & B

Scale bars are not needed for this Figure

- The Appendix figures and tables are missing the word 'Appendix' - see also guide to authors. Please correct callouts in manuscript text.

We added the word Appendix at all relevant places

- The list of authors in the reference list should be cut after 10 followed by et al.

We updated the reference list accordingly

- The legend to the movie should be zipped with is legend. Please also correct nomenclature - should be Movie EV1

Done

- We include a synopsis of the paper (see <http://emboj.embopress.org/>). Please provide me with a general summary statement and 3-5 bullet points that capture the key findings of the paper.

- We also need a summary figure for the synopsis. The size should be 550 wide by [200-400] high (pixels). You can also use something from the figures if that is easier.

We added a synopsis with a figure that summarizes our main findings on nuclear RT

- I have asked our publisher to do their pre-publication checks on the paper. They will send me the file within the next few days. Please wait to upload the revised version until you have received their comments.

We took into account all comments from the pre-publication checks

That should be all! Congratulations on a nice study

Thank you and EMBO Journal very much for your interest in our paper

With best wishes

Karin

Karin Dumstrei, PhD
Senior Editor
The EMBO Journal

Dear Christophe,

Thank you for submitting your revised manuscript to The EMBO Journal. I have now had a chance to take a look at the revised version and all looks good.

I am therefore very pleased to accept the manuscript for publication here.

best Karin

Karin Dumstrei, PhD
Senior Editor
The EMBO Journal

Please note that it is EMBO Journal policy for the transcript of the editorial process (containing referee reports and your response letter) to be published as an online supplement to each paper. If you do NOT want this, you will need to inform the Editorial Office via email immediately. More information is available here: https://emboj.embopress.org/about#Transparent_Process

Your manuscript will be processed for publication in the journal by EMBO Press. Manuscripts in the PDF and electronic editions of The EMBO Journal will be copy edited, and you will be provided with page proofs prior to publication. Please note that supplementary information is not included in the proofs.

Should you be planning a Press Release on your article, please get in contact with embojournal@wiley.com as early as possible, in order to coordinate publication and release dates.

If you have any questions, please do not hesitate to call or email the Editorial Office. Thank you for your contribution to The EMBO Journal.

EMBOD REFEC

YOU MUST COMPLETE ALL CELLS WITH A BRIEF BACKGROUND. It is PACE AFTER THAT THE CHECKLIST MUST BE DICTIONARY ALPHABETICALLY ORDERED

Corresponding Author Name: Francesca R Mounis, Christophe Zimmer
Journal Submitted to: EMBO Reports
Manuscript Number: EMBOR-2020-105247